# Zoonotic Mutation of Highly Pathogenic Avian Influenza H5N1 Virus Identified in the Brain of Multiple Wild Carnivore Species

**DOI:** 10.3390/pathogens12020168

**Published:** 2023-01-20

**Authors:** Sandra Vreman, Marja Kik, Evelien Germeraad, Rene Heutink, Frank Harders, Marcel Spierenburg, Marc Engelsma, Jolianne Rijks, Judith van den Brand, Nancy Beerens

**Affiliations:** 1Wageningen Bioveterinary Research, Wageningen University & Research, Lelystad 8221 RA, The Netherlands; evelien.germeraad@wur.nl (E.G.); rene.heutink@wur.nl (R.H.); frank.harders@wur.nl (F.H.); marc.engelsma@wur.nl (M.E.); 2Dutch Wildlife Health Centre, Utrecht University, Faculty of Veterinary Medicine, 3584 CL Utrecht, The Netherlands; m.kik@uu.nl (M.K.); j.m.rijks@uu.nl (J.R.); j.m.a.vandenbrand@uu.nl (J.v.d.B.); 3Division of Pathology, Faculty of Veterinary Medicine, Utrecht University, 3584 CL Utrecht, The Netherlands; 4NVWA Incident- and Crisiscentre (NVIC), Netherlands Food and Consumer Product Safety Authority, 3511 GG Utrecht, The Netherlands; m.a.h.spierenburg@nvwa.nl

**Keywords:** HPAI H5N1 influenza, carnivores, brain, virology, pathology, immunohistochemistry, wildlife, full genome sequences, phylogenetic analysis, zoonotic mutation

## Abstract

Wild carnivore species infected with highly pathogenic avian influenza (HPAI) virus subtype H5N1 during the 2021–2022 outbreak in the Netherlands included red fox (*Vulpes vulpes*), polecat (*Mustela putorius*), otter (*Lutra lutra*), and badger (*Meles meles*). Most of the animals were submitted for testing because they showed neurological signs. In this study, the HPAI H5N1 virus was detected by PCR and/or immunohistochemistry in 11 animals and was primarily present in brain tissue, often associated with a (meningo) encephalitis in the cerebrum. In contrast, the virus was rarely detected in the respiratory tract and intestinal tract and associated lesions were minimal. Full genome sequencing followed by phylogenetic analysis demonstrated that these carnivore viruses were related to viruses detected in wild birds in the Netherlands. The carnivore viruses themselves were not closely related, and the infected carnivores did not cluster geographically, suggesting that they were infected separately. The mutation PB2-E627K was identified in most carnivore virus genomes, providing evidence for mammalian adaptation. This study showed that brain samples should be included in wild life surveillance programs for the reliable detection of the HPAI H5N1 virus in mammals. Surveillance of the wild carnivore population and notification to the Veterinary Authority are important from a one-heath perspective, and instrumental to pandemic preparedness.

## 1. Introduction

The 2021–2022 highly pathogenic avian influenza (HPAI) season was the lengthiest so far in Europe, with more countries and poultry farms than ever before reporting infections. Other continents, such as North America, were also heavily affected [1,2]. The high number of HPAI infections in wild birds during this epizootic was unprecedented, and fatal infections occurred in a broad range of bird species [3]. The HPAI H5N1 viruses are classified as influenza A viruses, belonging to H5 clade 2.3.4.4b, and six different genotypes have been reported already in Europe during this epizootic [3]. Besides the detections in avian species, HPAI H5 virus infections were reported in mammals in several European countries (e.g., Finland, Sweden, Ireland, and Belgium), USA, Canada, and Japan. Many affected species belong to the order Carnivora [3,4,5], either terrestrial Carnivora such as the red fox (*Vulpes vulpes*), the American black bear (*Ursus americanus*) [3,6], or marine Carnivora such as seals. However, cases were also recently detected in marine Artiodactyla, specifically porpoises and dolphins [3]. Currently, a likely route of infection is through ingestion, e.g., feeding on infected birds or carcasses; however, in the case of the seals in North America, interspecies transmission could not be excluded [7].

In several wild carnivores, identified mutations in the virus genome were associated with mammalian adaptations. In the Netherlands, in the first three red foxes infected during the 2021–2022 outbreak, which presented neurological signs, the HPAI H5N1 virus was mainly detected in the brains, whereas low or no virus was detected in throat, nasal, or rectal swabs of the foxes. Genetic analysis identified mutation E627K in PB2 in two of the HPAI H5N1 viruses isolated from these animals. The PB2-627K mutation increased virus replication in mammalian cell lines [8]. This mutation is an adaptation of the virus polymerase protein, that likely stimulates virus replication in the lower body temperature environment in mammals compared to birds [9,10]. The PB2-627K mutation also increased HPAI pathogenicity in vitro and in vivo in mice models [11,12], but more mutations are required for efficient replication and transmission between mammals [13,14,15,16]. Currently, no mutations, associated with a switch to usage of the human receptor for virus entry, have been identified in HPAI viruses from avian or mammalian species. However, the detection of viruses carrying markers for mammalian adaptation, which are correlated with increased replication and virulence in mammals, may increase the risk for transmission to humans.

Continued in-depth analysis of mammal cases during an HPAI outbreak is therefore of paramount importance. This study characterized the HPAI H5N1 viruses detected in the period from December 2021 to February 2022 in red foxes and several other terrestrial carnivore species in the Netherlands and analyzed the associated pathology. Full genome sequencing of the viruses was performed to determine the relationship between the viruses found in these carnivores and wild birds, and to identify mutations associated with mammalian adaptation. In addition, virus pathogenesis was studied to identify the potential entry route of the virus, the sites of virus replication, and the associated lesions. Collectively, we showed that: (1) the carnivore viruses were less related to each other than to wild birds; (2) across species, the PB2-E627K mutation associated with mammalian adaptation was found as detected in foxes in the previous study [8]; (3) the brain of most mammals tested positive for viral RNA whereas limited viral RNA or protein was observed in other organs. We conclude that brain samples are important for reliable detection of the HPAI H5N1 virus in mammals, and therefore must be routinely included besides throat and anal swabs in the surveillance program, and HPAI infections in mammals must be made notifiable to the Veterinary Authority.

## 2. Materials and Methods

### 2.1. Pathology, Immunohistochemistry, and Virology Sampling

Wild carnivores that tested positive for HPAI H5N1 virus from December 2021 to February 2022 in the Netherlands, were included in this study. Cases were identified mainly through targeted surveillance of euthanized wild carnivores with ante-mortem neurological signs; such cases have been routinely submitted for HPAI and rabies testing since the end of 2021. Additional cases were identified through general wildlife disease surveillance on found dead wild carnivores: animals, which showed signs of viral infection during pathological examination were systematically tested for HPAI infection. Before necropsy, anal, and throat swabs were obtained for virological examination and after macroscopic evaluation, tissue samples were obtained for histopathology and immunohistochemistry (IHC) and fixed in 10% neutral buffered formalin. Brain tissue was sampled for virological examination. An overview of sampled material per animal is provided in Appendix A for virology and Appendix A for pathology. Swabs and tissue samples for virological examination were processed as described previously [8]. Tissue samples for histology were processed and embedded in paraffin and stained with hematoxylin and eosin (HE). Influenza A nucleoprotein expression by IHC in tissue samples was evaluated as described previously [17].

### 2.2. Virus Detection, Subtyping and Isolation

For virus detection, viral RNA was extracted using the MagNA Pure 96 system (Roche, Basel, Switzerland). The AI virus was detected by a quantitative real-time RT-PCR targeting the matrix gene (M-PCR) and detecting all influenza A viruses, as described previously [18]. For at least one sample of each animal the HA cleavage site sequence and the N subtype were determined by Sanger sequencing, as described previously [18]. For virus isolation, a swab or tissue suspension from each animal was inoculated into the allantoic cavity of 10-day-old embryonated specific pathogen-free (SPF) chicken eggs, according to standard protocols [19]. Isolated viruses were subtyped by the hemagglutination inhibition assay [19].

### 2.3. Virus Genome Sequencing and Phylogenetic Analysis

All virus genome sequences were determined directly on the swab or tissue samples. Virus RNA was purified using the High Pure Viral RNA kit (Roche, Basel, Switzerland), amplified using universal eight-segment primers and directly sequenced, as described previously [18]. Purified amplicons were sequenced at high coverage (average > 1000 per nucleotide position) using the Illumina DNA Prep method and Illumina MiSeq 150PE sequencing. The reads were mapped using the ViralProfiler-Workflow, an extension of the CLC Genomics Workbench (Qiagen, Hilden, Germany). Consensus sequences were generated by a reference-based method. Reads were first mapped to a reference set of genomes, and subsequently remapped to the closest reference sequence. Finally, the consensus sequence of the complete virus genome was extracted and minority variants were called using a cutoff of 1%. The consensus sequences of the viruses present in the mixed cerebrum and cerebellum samples were submitted to the GISAID EpiFlu-database.

In addition to the virus sequences obtained from carnivores and wild birds in the Netherlands in this study, H5N1 genome sequences of mammals from the period September 2021–October 2022 were retrieved from the GISAID database [20]; accession date 24 November 2022. Furthermore, a selection of related H5N1 sequences from wild birds in Eurasia was downloaded from the GISAID database and included in the phylogenetic analysis. Phylogenetic analysis of the complete genome sequences was performed for each genome segment separately: the viral sequences were aligned using MAFFT v7.475 [21] and the phylogeny was reconstructed using maximum likelihood (ML) analysis with IQ-TREE software v2.0.3 and 1000 bootstrap replicates [22]. The ML tree was visualized using the R package ggtree [23]. The contributors of the GISAID sequences used in the phylogenetic analysis are acknowledged in Appendix A.

## 3. Results

### 3.1. Virological Analysis of Infected Carnivores

During the period December 2021–February 2022, 21 wild carnivores suspected of HPAI H5N1 infection based on neurological signs or evidence of viral infection in post-mortem sections were submitted for testing. Of these, 14 animals tested positive for influenza A virus using the M-PCR. Three red foxes (*Vulpes vulpes*) were analyzed in a previous study [8]. The current study describes the analysis of the 11 other carnivores: red fox (*n* = 6), polecat (*Mustela putorius*) (*n* = 3), badger (*Meles meles*) (*n* = 1), and otter (Lutra lutra) (*n* = 1) (Table 1). The locations where the infected carnivores were found were dispersed over the Netherlands (Figure 1) and the cases did not cluster geographically. When examined, overall viral loads in brain tissues were higher than those in the swabs of the throat and intestinal tract. All viruses detected were of the subtype HPAI H5N1. Virus isolation was performed in eight animals, showing the presence of the infectious virus in five animals (Appendix A).

### 3.2. Phylogenetic and Genetic Analysis of Mammalian Viruses

Full genome sequencing was performed on samples derived from the 11 carnivores to study the genetic relationship between the viruses. Phylogenetic analysis showed that the carnivore viruses belonged to H5 clade 2.3.4.4b, and clustered with viruses found in wild birds during the HPAI H5N1 2021–2022 epizootic in the Netherlands (Figure 2, Appendix A). Similar viruses were also detected in wild birds and mammals in other European countries (Figure 2 and Appendix A). Two distinct clusters of the viruses were found during the 2021–2022 epizootic in birds and mammals, with different genetic constitutions (Appendix A). Viruses in one cluster had a genetic constitution similar to the HPAI H5N1 viruses detected in the previous 2020–2021 epizootic, whereas the viruses in the second cluster had obtained novel PB2, PA, and NP segments. The carnivore viruses were detected in both clusters, and there appears to be no correlation between the phylogenetic cluster and the animal species. The carnivore viruses were not closely related based on the phylogenetic analysis, and therefore these mammals were likely infected by independent introductions from wild birds.

Genetic analysis was performed to investigate whether mutations implicated in adaptation to mammalian hosts occurred in the virus genomes. Mutation screening identified the previously described E627K mutation in the PB2 segment in 8 of the 11 carnivores: 5 out of the 6 foxes, in 2 out of the 3 polecats, and in the otter (Table 1). No other mutations associated with adaptation to mammals were identified in the virus genomes (results not shown). A minority variant analysis of the next-generation sequencing data was performed on all carnivore brains, to study the composition of complete virus population in the brain. This analysis identified a mixture of the avian (627E) and mammalian (627K) PB2 variant in samples from fox (nr. 2). The avian 627E residue was still present in 5.3% of the virus population in the brain sample (Appendix A). These results suggest that the 627K variant likely emerged within this animal from the avian variant. For the other 10 carnivores, no minority variants were detected at this position in the virus genomes.

### 3.3. Pathology and Virus Expression

The included animals were all (young) adults with a body condition, based on fat and muscle development, ranging from poor to good (Appendix A). All animals showed a non-suppurative meningitis, encephalitis, or meningoencephalitis with variation in severity from mild to severe and most prominent in the cerebrum (Appendix A). In severe cases, such as the otter (nr. 10), large parts of the brain were affected, characterized by extensive gliosis, perivascular cuffing, neuronal degeneration, and necrosis with the presence of cellular debris (Figure 3A), hemorrhages, and meningitis. Severely affected parts of the cerebrum showed only minimal virus expression (Figure 3A–D), while in parts with milder changes (mainly gliosis), there was extensive viral expression in the neurons (Figure 3E,F). The olfactory bulb, a possible route of viral entry, was also investigated in five animals. However, no clear histopathologic changes nor viral expression was observed in this part of the central nervous system. The morphology of the brain lesions and cell type with virus expression did not differ among animal species. Only one fox (nr. 3) showed a more suppurative meningoencephalitis with intralesional accumulation of bacteria, most likely consistent with a secondary bacterial infection (Appendix A).

Besides the brain changes, macroscopically, the poorly collapsed, red or marbled red lungs were also a common finding occurring in 7 out of 11 animals (Appendix A). Most of these animals showed in the lungs a nematode infestation (in red foxes consistent with *Angiostrongulus vasorum*) with a variable-associated subacute to chronic pneumonia. Overall, these lung changes were not associated with viral protein expression. Only two animals (fox (nr. 2) and polecat (nr. 9)) showed minimal viral expression in a few mononuclear cells (epithelial cells or macrophages) in the lungs. In the upper respiratory tract (nasal conchae and trachea) showed the changes were mild with only virus expression in the respiratory and olfactory epithelial cells in the nose of one polecat. Parasites (most consistent with *Capillaria spp.*) were observed in the nasal conchae of one fox (nr. 4) and one polecat (nr. 8). From all animals, except for two foxes, extra-respiratory tissues were evaluated. In the intestinal tract, we observed no significant histopathologic changes nor viral expression. Variable histopathologic changes were observed in kidney, liver, or heart (Appendix A). These were not associated with viral expression, except in the otter (nr. 10). This animal showed minimal viral expression in epithelial cells and mononuclear cells in the pancreas, which was associated with a mild lymphoplasmacytic infiltrate and necrosis.

## 4. Discussion

The HPAI H5N1 epizootic in 2021–2022 is the largest that occurred in Europe so far and was associated with massive mortality amongst wild birds. This study showed that at least four different wild carnivore species became infected with HPAI H5N1 viruses in the Netherlands. Within a three-month period, 14 out of 21 carnivores suspected of HPAI H5N1 infection based on ante-mortem behavior or post-mortem indications for viral infection, tested positive for the virus. The virus was detected in nine red foxes, three polecats, one otter and one badger. This study confirmed the initial findings in three of the red foxes [8] by greater numbers of animals and across four species. Higher viral RNA loads were consistently detected in the brains of the animals studied compared to those detected in throat or anal swabs. Interestingly, 3 out of 11 animals in this study were negative in anal and throat swab and HPAI diagnosis would have been missed without additional brain samples. Other studies also reported the presence of HPAI H5 virus in the brain of various wild carnivore species [3,5,7]. Full genome sequencing of the carnivore viruses, followed by phylogenetic analysis, demonstrated that they belonged to H5N1 clade 2.3.4.4b and that the carnivore viruses were related to viruses detected in wild birds in the Netherlands. Consistent with the results from the initial study [8], the carnivore viruses were not genetically closely related among themselves and they did not cluster geographically. This suggests that the animals were infected by separate virus introductions likely originating from wild birds. There was no evidence for mammal to mammal spread within this small number of cases. However, there was no active surveillance for the virus in ambulatory or mildly infected mammals without clinical signs, which may also spread the virus. Samples from wildlife harvested for hunting purposes, wildlife culled as a management tool, or culled for intervention of disease control, could be used for that type of surveillance.

Previous studies showed that (wild) carnivores were at risk for HPAI infection [25,26]. Reperant et al. demonstrated in an experimental infection that red foxes were susceptible to HPAI H5N1 clade 2.2 after eating infected bird carcasses. Virus excretion was seen from 3 to 5 days after infection with only mild or even no pneumonia and without brain lesions [27]. However, they used a different HPAI clade than we observed in our mammals and these experimental infected foxes were euthanized 7 days after feeding carcasses and perhaps more time was needed to develop H5N1-induced brain lesions as observed in our cases after natural infection. Another study showed that cats were also highly susceptible to HPAI H5N1 (A/Vietnam/1194/2004) infection after eating virus-infected chicks and demonstrated systemic disease including brain lesions already 7 days after experimental infection [28]. These findings lead us to the hypothesis that the red foxes, polecats, badger, and otter were infected by eating carcasses; however, ingesting water contaminated with infected bird feces could also be a route of infection [29,30].

One of the aims of this study was to find pathological evidence for the route of viral entry. This may be important for outcome of disease, as was seen in cats 7 days after experimental infection with HPAI H5N1 via different infection routes: intratracheal, feeding on virus-infected chicks, and horizontal transmission [28]. All infection routes induced respiratory signs with the associated virus and lesions. However, only the cats infected via ingestion of infected chicks or, intratracheally infected cats showed brain lesions, not the horizontally infected cats. Additionally, a virus-associated ganglion neuritis in the nervous plexi of the small intestine was only found in cats fed virus-infected chicks [27]. This ganglion neuritis was suggestive for HPAI infection via the nervous tract or intestinal lumen. Ganglion neuritis and other HPAI-associated intestinal changes were not observed in our wild carnivores; however, we cannot fully exclude this, because autolysis from field samples may have hampered the observation of these changes. It is not only the respiratory route and the intestinal route that may be involved in infection of carnivores with HPAI H5N1 viruses. In ferrets, after experimental intranasal infection, the virus entered the brain via the olfactory mucosa from which it spreads to the olfactory bulb and the rest of the central nervous system [31]. In this study, we investigated the nasal conchae and olfactory bulb for a limited number of wild carnivores. Virus expression was observed in the nasal conchae of only one polecat, with absence of clear histopathologic changes and there was no virus expression in the olfactory bulb. Therefore, in this study, no support was found for the olfactory route of entry. However, we did not follow the animals over time and mostly sampled at the end stage of disease. Thus, we cannot draw a conclusion on the viral entry route of the brain infection, and the respiratory, olfactory, and intestinal routes may have altogether contributed to the HPAI infection in the brain.

Brain lesions of experimental [27,28] and natural infected carnivores [5] were characterized by the accumulation of glial cells, perivascular mononuclear cuffs, and scattered and neuronal necrosis in the parenchyma, leptomeninges, and choroid plexi. The carnivores described in our study showed similar brain lesions. These varied in severity, probably due to the timing and dose of the field infection, that were not known. In the cases investigated in this study, the non-suppurative meningitis and/or encephalitis was most prominent in the cerebrum, but also present in some cases in the cerebellum. Advice is therefore to sample both parts of the brain for virus detection. Lesions in the cerebellum may cause more severe neurological signs related to coordination imbalance and gait disorders, such as ataxia, when compared to lesions in the cerebrum [32]. Cerebellar lesions make it difficult for an infected animal to find and catch their food and to walk or run away when threatened. Cerebellar lesions in our cases could be associated with a longer duration of infection as was demonstrated in intranasally infected ferrets, where the cerebrum was positive for antigen at 5 days post infection (dpi) while the cerebellum later became positive at 7 dpi [31]. Viral protein was clearly expressed in neurons, but also in glial cells and mononuclear cells in meninges. In severely affected areas with extensive loss of neurons and presence of cellular debris, there is only limited virus expression due to the lack of viable cells. This suggests that virus protein expression with IHC was below the detection limit of this method, and that RT-PCR is needed to detect viral RNA in the brain at the end stage of disease.

It is currently unclear which factors have contributed to the increased number of infections detected in wild carnivores. The HPAI H5N1 clade 2.3.4.4b virus may be more infectious for mammals with more prominent progression to encephalitis, which made animals with neurologic signs easier to detect or there may be just a higher prevalence of the virus in wild birds during the 2021–2022 epizootic compared to previous epizootics. The fact that we detected HPAI H5N1 infections in different carnivore species suggests that other mammalian species, and domestic animals such as cats and dogs, are at risk of becoming infected by feeding on sick or dead wild birds. Viral protein expression was only found in a few animals in the respiratory tract and not in the intestinal tract. This indicates that at the time point when these animals died, there was only limited virus shedding. It cannot be excluded that infected mammals have shed the virus at an earlier time point in the infection. However, in our study we found no evidence for virus transmission between animals based on phylogenetic analysis of the viruses.

Genetic analysis of the carnivore viruses in this study identified the zoonotic mutation PB2-E627K in 8 out of 11 cases. The fact that this mutation was not detected in any of the wild bird sequences during the 2021–2022 epizootic in the Netherlands suggests this mutation quickly arises upon infection of mammals. In one fox (nr. 2), the avian E627-variant was still present in the viral genome as a minority population, further supporting the emergence of the mutation within this mammal. A previous analysis of two fox viruses also supported that the mutation arose after infection of the mammals, as minority variants were detected at position PB2-627 [8]. Mutation E627K is likely an adaptation to the lower body temperature in the mammalian upper respiratory tract compared to that of avian species. We previously showed that this mutation increases the replication of the HPAI H5N1 virus in mammalian cell lines at lower temperatures [8]. The fact that the mammalian adaptation marker E627K was found in many of the carnivore viruses suggests that this virus can rapidly adapt to replication in mammals. However, previous research has indicated that a combination of viral adaptations is required for efficient air-borne transmission of HPAI viruses between mammals [13]. Although the chance that such mutations will arise in an infected animal is very small, the impact of the emergence of a zoonotic virus with potentially pandemic characteristics may be large.

In this observational wildlife study, carnivore carcasses were sampled for different surveillance purposes and therefore carcasses showed variation in degree of autolysis and also different sampling strategies were applied. In the future, a more universal wildlife sampling approach within different institutes will enable structural analysis with more reliable results. Another important caveat of sampling found dead or euthanized wild animals compared to an experimental infection is the unknown time point of infection, which hampers the evaluation of the HPAI H5N1 pathogenesis. Finally, many of the wild carnivores were infested with lung parasites [33]. Especially in red foxes, the presence of *A. vasorum* larvae in the lungs was a common finding [34] and may have overshadowed changes due to HPAI lung pathology. Furthermore, the lung pathology and the clinical relevance of these parasites is variable in wildlife and it is unclear if these co-infections influenced susceptibility to HPAI. Besides these limitations, our results can be used to improve HPAI wildlife surveillance.

## 5. Conclusions

Our study showed that brain samples were needed for the reliable detection of the HPAI H5N1 virus in mammals, and therefore must be routinely included, besides throat and anal swabs, in surveillance programs. HPAI viruses were detected in multiple carnivore species, frequently with mutations indicative of mammalian adaptation. Therefore, detection of HPAI viruses in mammals must be made notifiable to the Veterinary Authorities. Surveillance of the wild carnivore population is important from a one heath perspective, and instrumental to pandemic preparedness.

## Figures and Tables

**Figure 1 pathogens-12-00168-f001:**
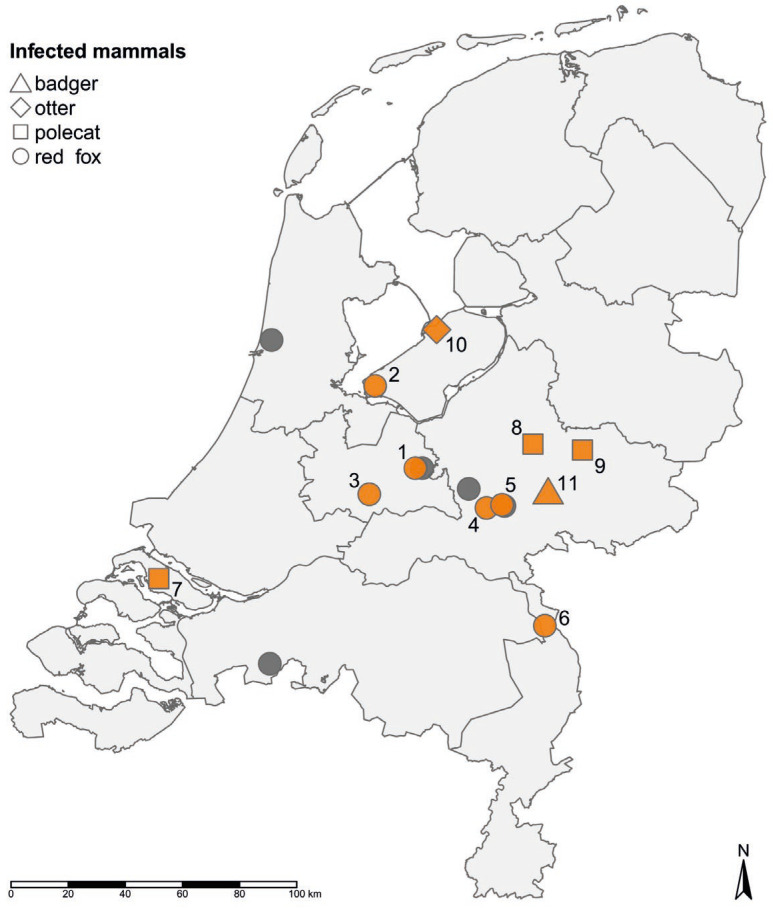
Map of the locations were HPAI H5N1 infected carnivores were found during the 2021–2022 epizootic in the Netherlands. The infected carnivores analyzed in this study are marked in orange, and numbers are corresponding to Table 1. The infected mammals analyzed in previous studies [8] are depicted in grey. The map was generated using the R software package tmap [24].

**Figure 2 pathogens-12-00168-f002:**
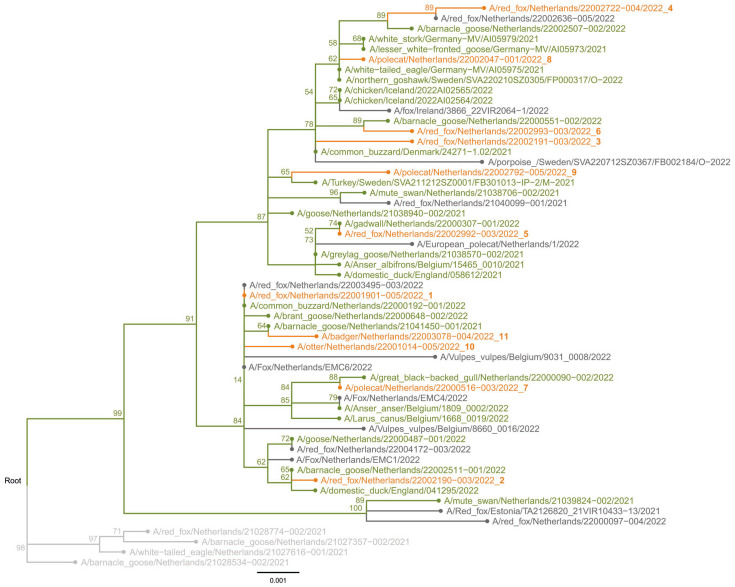
Phylogenetic tree of the HA segment obtained with the maximum likelihood method showing the viruses detected in the carnivores from the current study (orange). The previous carnivore H5N1 virus sequences from the period 2021–2022 (dark grey), related sequences from wild birds (green), and relevant H5N1 sequences from the previous 2020–2021 avian influenza season (light grey) are also shown. Bootstrap values above 50% are shown at the nodes.

**Figure 3 pathogens-12-00168-f003:**
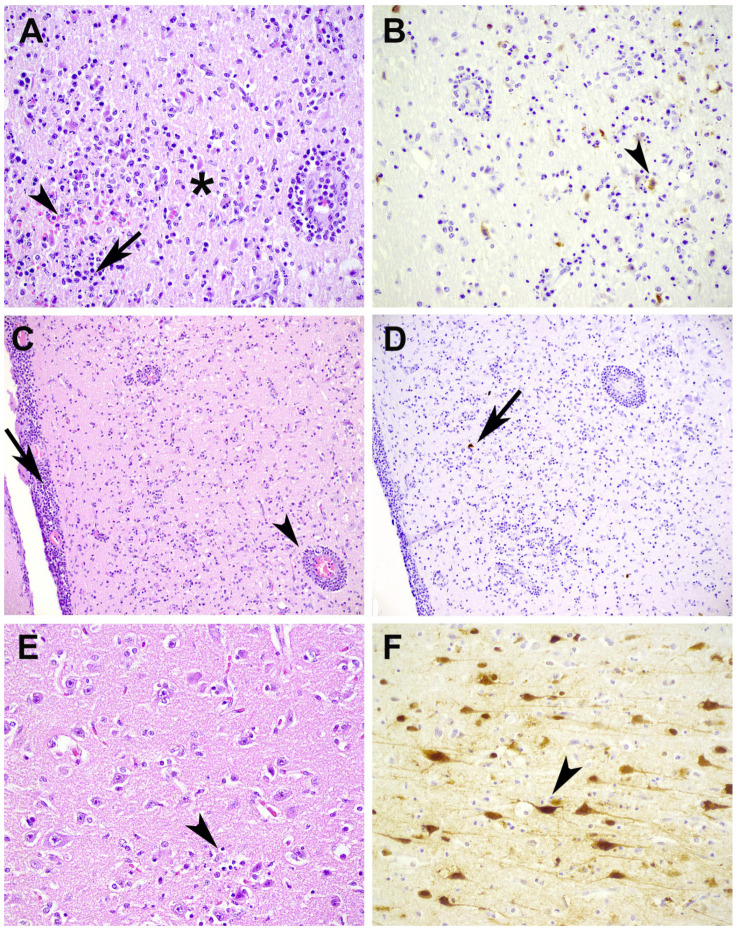
Brain histopathology and virus expression in the cerebrum of an otter infected with HPAI H5N1. (**A**) Severe changes in the white matter with loss of neurons, neuronal degeneration (asterisk), cellular debris (arrow), gliosis, and hemorrhages (arrowhead). Hematoxylin and eosin stain (HE), objective 40×. (**B**) Virus expression in neurons (arrowhead), immunohistochemistry (IHC) influenza A protein, objective 40×. (**C**) Perivascular cuffing (arrowhead) and meningitis (arrow). HE, objective 40×. (**D**) Only minimal virus expression in a few degenerated neurons (arrow), IHC, objective 20×. (**E**) Cerebrum with mild to moderate gliosis in neuropil and surrounding neurons (arrowhead), HE, objective 40×. (**F**) Clear virus expression in neurons (arrowhead), IHC, objective 40×.

**Table 1 pathogens-12-00168-t001:** Overview wild carnivores and Influenza infection.

Case nr	Species	Neurologic Signs	RT-PCR (Ct Value)		Mutation	IHC (Influenza A Protein)			
			Throat Swab	Brain Tissue	E627K (Brain)	Brain	Resp	GIT	Other
1	Fox	yes	30.75	26.35	yes	−	−	−	−
2		found dead	28.36	15.95	yes	+	+	−	−
3		found dead	28.88	17.64	yes	+	na	−	−
4		found dead	24.38	16.22	yes	+	−	−	−
5		found dead	24.79	18.66	no	+	na	na	na
6		yes	39.21	20.01	yes	+	na	na	na
7	Polecat	yes	No Ct	23.22	no	−	−	−	−
8		yes	22.62	na	yes*	−	+	−	−
9		found dead	17.98	16.94	yes	−	+	−	−
10	Otter	yes	34.61	18.54	yes	+	−	−	+
11	Badger	yes	No Ct	28.83	no	−	−	−	−

Real-time RT-PCR targeting the influenza matrix gene; * sample from throat swab; na: not analyzed; IHC: immunohistochemistry, positive staining (+), and no staining (−); brain: cerebellum and cerebrum; resp: respiratory; GIT: gastro-intestinal tract.

## Data Availability

The virus genome sequences generated in this study were submitted to the GISAID database.

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
