# Peer review of "Zoonotic Mutation of Highly Pathogenic Avian Influenza H5N1 Virus Identified in the Brain of Multiple Wild Carnivore Species"

_pathogens, 2023, doi:10.3390/pathogens12020168_

Round 1

Reviewer 1 Report

This paper studied samples from 11 severely ill or dead mammals from four different species in the Netherlands in 2021-2022 for the presences of avian influenza A(H5N1) virus in various samples using PCR and immunohistochemistry.  Viruses were sequenced and compared to others in GIDAID.  The results showed that the viruses across the 11 mammals did not cluster and were less similar to each other than to those found in wild birds that year.  A mutation (E627K) was found in all but 3 mammals and in at least one mammal from each species.  This mutation has been shown to be associated with mammalian adaptation.  Finally, viral RNA was mostly likely to be found in brain tissue samples rather than respiratory samples, which is likely related to the fact that these mammals suffered from a meningo-encephalitis type illness.  These findings, which were clearly presented, help us better understand the current epizootic of H5N1 clade 2.3.4.4b and the risk of virus spread to mammals, and provide important suggestions for improving future surveillance in mammals (i.e., best sample type). 

Specific comments

Abstract.  I understand that this is an unstructured abstract, but it is not clear what is published data (old) and what is new (from this study).  Can you maybe make that distinction and perhaps add the number of mammals studies in this current study, which I think was 11?

Line 21.  You should also include the type, A.

Line 44.  Same here.  I would say “…A(H5N1) viruses…”

Line 56.  There is an extra space before period at the end of the sentence.

Lien 62, There is an extra space before the comma.

Line 63.  The word temperature should be singular.

Line 67-69.  I think there is an error in the grammar.  Perhaps you meant to say “…for mammalian adaptation which hare correlated…”

Line 74-84.  This is really methods and I suggest moving it.

Line 92.  Change to “the Veterinary Authority.”

Line 161.  I find this section a little confusing.  On lines 166-167 you say that viruses in one cluster matched the epizootic (which I think here you mean wild birds).  But then on line 171-173, you say viruses in the other cluster matched wild birds too.  I think you could reorganize it to say there were two clusters in the mammals, both of which were also founds in the wild birds (if that if the correct interpretation).

Line 247.  Suggest changing to read “…would have been missed…”

Line 255.  While I suppose this sentence is technically true, five of your mammals were found dead when collected and the others were probably so sick that they were alone.  By the nature of your study, you don’t know whether there was onward spread or not.  I suggest you re-write that sentence to say something like this, “There was no evidence for mammal to mammal spread within this small sample, but we did not search for the virus in ambulatory mammals.”

Line 260.  I suggest modifying this sentence to say “…different HPAI clade than we observed in our mammals…”

Line 265-266.  Instead of saying “By this…” I suggest you say, “These findings lead us to hypothesis that the red foxes…”

Line 271-272.  You might consider using standard terms for transmission routes (direct, aerosols, ingestion, indirect) then using your terminology as an example of these.  For example, ingestion (e.g., feeding on virus-infected chicks).  I think horizonal is direct since in that paper the cats were put in the same cage.

Lines 313-316.  Something that is also unclear and not answered by this study, but of importance and perhaps worth mentioning, is whether the virus only causes severe/fatal disease in infected mammals.  If not, there could be a bunch of mildly ill infected mammals running around and spreading the infection.  This has important public health consequences so may be worth a sentence somewhere.

Lines 326-327.  Here again I am confused.  Doesn’t this contradict what you said in lines 171-172?  “HPAI H5N1 viruses with both genetic constitutions were also found in wild birds during this epizootic in the Netherlands…”

Line 344-346.  You could also mention that during times of large epidemics in birds that it might be prudent to also sample of live mammals to understand the range of infection in mammals.

Line 348.  Need space after H5N1 and before pathogenesis.

Reviewer 2 Report

It is very pleasant to read a well-written and clearly illustrated article, that presents a perfectly carried study out.

Authors obtained new data concerning spread and detection of the HPAI H5N1 virus in wild carnivores. They concluded that brain samples are important for reliable detection of the HPAI H5N1 virus in mammals. Therefore, they recommended to include brain samples in the surveillance program.

Thank authors for excellent work.

 There are extremely small remarks.

The word ‘polecat’ is sometimes written as ‘pole cat’.

Line 18: twice repeating ‘Correspondence:’

Line 370 – ‘writ-ing’ should be without hyphen
